# Availability and Quality of School Facilities as a Determinant of Local Economic Development: The Slovak Experience

**Viera Papcunová** [1,2], **Marek Dvořák** [3], **Roman Vavrek** [4,*], **Jarmila Mižičková** [5], **Petra Harasimová** [4], **Monika Víchová** [4] and **Tomáš Váňa** [4]

1    Institute of Economics and Management, Faculty of Natural Sciences and Informatics, Constantine the Philosopher University in Nitra, 949 74 Nitra, Slovakia

2    Department of Regional Economics and Administration, Faculty of Economics and Administration, Masaryk University, 602 00 Brno, Czech Republic

3    Department of Trade and Finance, Faculty of Economics and Management, Czech University of Life Sciences Prague, 165 00 Prague, Czech Republic

4    Department of Public Economics, Faculty of Economics, VŠB—Technical University of Ostrava, 702 00 Ostrava, Czech Republic

5    Department of Management, Faculty of Management and Business, University of Prešov, 080 01 Prešov, Slovakia

*    Correspondence: roman.vavrek@vsb.cz

**Abstract:** The availability and quality of school facilities has long been a hot topic, not only at the level of self-government, but also in respect of the whole public administration within the context of Slovakia. COVID-19 and the current military conflict in Ukraine are jointly putting increased pressure on local self-governments as the founders of kindergartens and primary schools. Whether the availability and quality of school facilities is a determinant of local development is a question for which there are very few studies. The aim of this paper is to identify long-term satisfaction with the availability and quality of kindergartens and primary schools in the Červený Kameň microregion; this is understood in the context of these institutions serving as an important determinant of local development. In this paper, a qualitative evaluation (survey) of parents of students in the Čerevený Kameň microregion was conducted. In order to increase the objectivity of the obtained results, the survey was conducted two times with an interval of 7 years (2014 vs. 2021). Our results show that parents perceive the availability and quality in the same way even over a long period, but without confirmed statistically significant differences. At the same time, however, we found that satisfaction with the analyzed school facilities changed over time.

**Keywords:** local development; local self-government; performance; facilities availability; schools

## 1. Introduction

The educational level of the population is an important endogenous factor of local development, which is closely related to economic development. Therefore, development is often understood in conjunction with socio-economic development, i.e., in the context of economic and social expression (Fáziková and Belajová 2005). Grenčíková and Španková (2012), as well as Beresecka (2014), confirm this notion. Further, they also note that both economic and non-economic perspectives on education demonstrate that education is part of the standard of living in respect of the population. Moreover, it becomes an essential need of the people and is a basic determinant of the economic development of the region. However, the success of the socio-economic development of the territory largely depends on what prerequisites and dispositions the residents have in regard to their own work performance, as well as what space and opportunities they possess in respect of initiative and self-realization. The existence of school facilities in the territory thus directly affects the quality of human capital, which is an important endogenous factor of development

(Wokoun et al. 2008). Improving socio-economic development at the local level is a goal of the local interest in the context of supporting the creation of social changes in the area of the local economy (de Melo et al. 2020). Social change at the local level is closely related to the structure of education and, ultimately, subsequent entrepreneurial innovation and creative leadership approaches often result from pressure by citizens on local governments in order to address community issues (Watson and Hassett 2003). Therefore, development must be seen as a structural change due to the fact that it enables changes in economic, social, political, and institutional structures within the territory (de Souza 2012). This is also confirmed by Nurvita Dwi et al. (2022), who advise that improving people's quality of life is inseparable from the government's role in implementing policies and programs related to improving the quality of life in the community through government spending on education. Economists refer to education as "human capital" and education is considered to be a factor that is useful to both the individual and society. The basis of educational policy in most developed countries of the world is the perception of education and upbringing as a necessary condition not only for economic growth, but also for the modernization of politics and the individual development of individuals (Miňová 2013). The local self-government is responsible for pre-primary and primary education in cooperation with the state. This is due to the fact that a significant part of the funds for the operation of these facilities comes from the state budget. In all EU-27 countries—with the exception of Denmark, Latvia, and Sweden—within the time period of 2012–2019, public expenditure on pre-primary education did not exceed 1% of GDP. On the contrary, a different situation arose in the financing of primary education. Bulgaria, Czechia, Germany, Hungary, Austria, Romania, and Slovakia also spent less than 1% of GDP on primary education during the time period of 2012–2019. It was only during the period of 2013–2015 and in 2019 that Italy alone spent more than 1% of public expenditure on primary education; in other years, it was less than 1%. In Lithuania, in 2012, public spending on primary education was 0.74% of GDP, but in the following year this spending rose to the level of 2.43%. However, the country failed to maintain this trend, and in the following years public expenditure on primary education fell below 1% of GDP (Table 1). The amount of public expenditure on individual levels of education is closely related to the organizational provision of educational services in the country. As part of the reform of public administration in most countries, pre-primary and primary education were developed by local self-governments through their competencies, but methodological management remained with the ministries of education and financing was realized through government transfers.

Such "mixed" management and financing of education brings with it another problem, which is the evaluation of the quality of education. This is related to the fact that the field of education is part of public administration; as such, it is, therefore, difficult to quantify the outputs completely precisely due to the fact that it is a public service. This is also confirmed by Stejskal et al. (2017), who add that the position of the provider affects the quality of individual processes in the context of public administration within the hierarchy of public administration and its organizations. However, in the case of local self-governments, there are no suitable mechanisms that would signal, in advance, the insufficient quality and scope of the provided public services. According to Rýdl (2001), the prerequisite for effective assessment processes in this area is found within the definition of the term quality of education. According to the author, quality is defined not only by the internal needs of the educational system as a result of efforts and expectations in terms of goals at various levels (i.e., students, class, school, school system, educational system, and society), but also increasingly by external influences—such as the social organization and economic possibilities of the country. The quality of educational processes, institutions, and educational systems represents the desired level of functioning or production of these processes or institutions, which can be objectively measured and evaluated (Průcha 2000). Indicators of the quality of education suggest certain information regarding the current situation, as well as the current state of the school system and the processes that take place within this system. Based on these indicators, it is possible to evaluate the state of the

given system as satisfactory or unsatisfactory, and also whether the system is suitable for modification. The quality of education can be assessed primarily through qualitative indicators, due to the fact that it is a service, but also through quantitative indicators. Qualitative indicators consist of expressing the feelings and attitudes of pupils and teachers toward the conditions and situations at school, etc. Quantitative indicators include data that can most often be expressed in terms of numbers—for example, in this case, the percentage of people with a university degree, the success rate of graduations, financial costs per student, the number of qualified teachers, etc. (Janoušková and Maršák 2008; Cabelkova et al. 2021). According to Lynch (2015), the main indicator of school quality should be the performance of students—i.e., their educational results and their success—as this is the main goal of school institutions. Another important indicator is the social characteristics of students—e.g., their personal growth. A high-quality school has several characteristics, such as high-quality management; the high expectations of students and teachers; the continuous verification of knowledge; and the ensured quality of the organization by the school. In addition, it is also the size of the school that is important. Smaller schools have better student success rates. Furthermore, students in smaller schools, and therefore those in smaller groups, have better relationships with their classmates and teachers. Bhattacharjee (2019) provides us with several examples regarding this topic, which, in turn, can be an obstacle for quality education in schools. These barriers include: a lack of quality teachers; a lack of space in classrooms; a lack of specialist classrooms with the necessary equipment; a selection of the wrong teaching methodologies; inflexible curricula (curriculum should be flexible enough to adapt to both students and society); irregular meetings with parents; and lack of funds. Other examples can also include: an unbalanced number or ratio of students to teachers, incorrect teaching aids, or out-of-date teaching methods.

**Table 1.** Public expenditure on pre-primary education and primary education as % of GDP in the EU-27.

| | 2012 | | 2013 | | 2014 | | 2015 | | 2016 | | 2017 | | 2018 | | 2019 | |
|---|---|---|---|---|---|---|---|---|---|---|---|---|---|---|---|---|
| | PPE | PE | PPE | PE | PPE | PE | PPE | PE | PPE | PE | PPE | PE | PPE | PE | PPE | PE |
| Belgium | 0.68 | 1.56 | 0.71 | 1.55 | 0.70 | 1.53 | 0.70 | 1.54 | 0.70 | 1.54 | 0.68 | 1.51 | 0.69 | 1.52 | 0.68 | 1.49 |
| Bulgaria | 0.89 | 0.69 | 1.03 | 0.80 | 1.05 | 0.83 | 0.97 | 0.79 | 0.95 | 0.81 | 0.94 | 0.81 | 0.92 | 0.82 | 0.93 | 0.83 |
| Czechia | 0.54 | 0.75 | 0.54 | 0.73 | 0.55 | 0.74 | 0.52 | 0.77 | 0.49 | 0.74 | 0.53 | 0.81 | 0.55 | 0.85 | 0.58 | 0.91 |
| Denmark | 1.25 | 2.23 | : | 2.11 | : | 2.08 | : | : | 0.44 | 1.71 | 0.44 | 1.67 | 0.43 | 1.63 | 0.45 | 1.59 |
| Germany | 0.43 | 0.63 | 0.44 | 0.63 | 0.44 | 0.63 | 0.46 | 0.62 | 0.47 | 0.63 | 0.49 | 0.63 | 0.51 | 0.65 | 0.55 | 0.68 |
| Estonia | 0.43 | 1.27 | 0.35 | 1.49 | : | 1.21 | : | 1.25 | : | 1.29 | : | 1.36 | : | 1.46 | : | 1.61 |
| Ireland | 0.11 | 2.27 | 0.11 | 1.93 | 0.10 | 1.79 | 0.07 | 1.39 | : | 1.38 | : | 1.21 | : | 1.16 | : | 1.14 |
| Greece | 0.24 | 1.10 | 0.25 | 1.14 | 0.27 | 1.21 | 0.26 | 1.22 | 0.27 | 1.21 | 0.25 | 1.18 | 0.26 | 1.25 | 0.27 | 1.30 |
| Spain | 0.50 | 1.15 | 0.48 | 1.13 | 0.48 | 1.13 | 0.47 | 1.14 | 0.46 | 1.14 | 0.44 | 1.13 | 0.44 | 1.12 | 0.43 | 1.12 |
| France | 0.65 | 1.13 | 0.70 | 1.14 | 0.70 | 1.14 | 0.70 | 1.13 | 0.69 | 1.12 | 0.70 | 1.16 | 0.69 | 1.16 | 0.67 | 1.17 |
| Croatia | : | : | : | : | : | : | : | : | 0.52 | 1.64 | 0.54 | 1.65 | 0.55 | 1.68 | 0.56 | 1.66 |
| Italy | 0.45 | 0.98 | 0.45 | 1.02 | 0.46 | 1.00 | 0.48 | 1.02 | 0.44 | 0.91 | 0.47 | 0.98 | 0.48 | 0.96 | 0.49 | 1.02 |
| Cyprus | 0.34 | 1.98 | 0.35 | 1.97 | 0.35 | 2.01 | 0.35 | 2.05 | 0.33 | 2.06 | 0.32 | 1.88 | 0.30 | 1.90 | 0.28 | 1.86 |
| Latvia | 1.13 | 1.86 | 0.82 | 1.47 | 0.86 | 1.59 | 0.81 | 1.60 | 0.80 | 1.51 | 0.82 | 1.35 | 0.76 | 1.28 | 0.81 | 1.29 |
| Lithuania | 0.63 | 0.74 | : | 2.43 | 0.50 | 0.68 | 0.56 | 0.70 | 0.63 | 0.76 | 0.62 | 0.74 | 0.69 | 0.72 | 0.70 | 0.75 |
| Luxembourg | 0.65 | 1.43 | 0.53 | 1.16 | 0.57 | 1.23 | 0.57 | 1.23 | 0.47 | 1.02 | 0.48 | 1.05 | 0.48 | 1.04 | 0.52 | 1.14 |
| Hungary | 0.65 | 0.75 | 0.65 | 0.87 | : | 0.58 | 0.78 | 0.76 | 0.77 | 0.76 | 0.70 | 0.66 | 0.67 | 0.64 | 0.67 | 0.79 |
| Malta | 0.41 | 1.04 | 0.43 | 1.09 | 0.40 | 1.03 | 0.37 | 0.97 | 0.37 | 0.94 | 0.34 | 0.92 | 0.35 | 1.00 | 0.37 | 1.00 |
| The Netherlands | 0.41 | 1.40 | 0.38 | 1.31 | 0.37 | 1.24 | 0.36 | 1.21 | 0.35 | 1.16 | 0.34 | 1.15 | 0.35 | 1.16 | 0.35 | 1.15 |
| Austria | 0.47 | 0.88 | 0.47 | 0.91 | 0.48 | 0.90 | 0.49 | 0.92 | 0.50 | 0.91 | 0.51 | 0.91 | 0.49 | 0.89 | 0.36 | 0.86 |
| Poland | 0.56 | 1.50 | 0.59 | 1.50 | 0.61 | 1.54 | 0.61 | 1.49 | 0.64 | 1.47 | 0.67 | 1.44 | 0.70 | 1.46 | 0.73 | 1.25 |
| Portugal | 0.39 | 1.44 | 0.42 | 1.57 | 0.42 | 1.53 | 0.39 | 1.42 | 0.38 | 1.38 | 0.40 | 1.48 | 0.36 | 1.40 | 0.35 | 1.38 |
| Romania | 0.29 | 0.48 | 0.30 | 0.50 | 0.34 | 0.43 | 0.33 | 0.41 | 0.31 | 0.37 | 0.31 | 0.38 | 0.33 | 0.40 | 0.37 | 0.45 |
| Slovenia | 0.68 | 1.60 | 0.61 | 1.53 | 0.65 | 1.51 | 0.56 | 1.40 | 0.52 | 1.38 | 0.52 | 1.37 | 0.51 | 1.43 | 0.51 | 1.41 |
| Slovakia | : | : | 0.44 | 0.80 | 0.49 | 0.81 | 0.49 | 0.89 | 0.50 | 0.87 | 0.53 | 0.88 | 0.55 | 0.91 | : | : |
| Finland | 0.78 | 1.37 | 0.77 | 1.33 | 0.77 | 1.37 | 0.77 | 1.44 | 0.76 | 1.41 | 0.71 | 1.34 | 0.74 | 1.35 | 0.77 | 1.41 |
| Sweden | 1.29 | 1.75 | 1.29 | 1.73 | 1.30 | 1.75 | 1.29 | 1.77 | 1.29 | 1.85 | 1.21 | 1.89 | 1.21 | 1.92 | 1.12 | 1.94 |

Legend—PPE: pre-primary education; PE: primary education. : Unavailable data, Source: Eurostat, own processing.

In the last decade, most countries introduced a system of educational accountability. These accountability systems provide insight into the educational performance of educational institutions and are often used in order to inform governments, students, and parents. Most systems of educational accountability use either the percentages of the students who are passing exams and/or their average test or exam scores as indicators of the school's quality. In general, quality indicators can be divided into two categories: namely, performance indicators that are based on test or exam scores or performance indicators that are

based on passing or failing an exam and repeating a grade. Examples of systems of educational accountability that focus strongly on student performance can be found in the Dutch, English, and Belgian systems (Timmermans et al. 2014). Value-added modeling (VAM) is the most widespread evaluation tool that evaluates student outcomes in relation to teacher effectiveness; on this note, it has been applied primarily in Great Britain and the USA. VAM is a common name for several statistical procedures that attempt to link or establish causality between teacher performance and student scores on standardized tests (Hill 2009). A value-added model is a quasi-experimental statistical model that provides estimates regarding the contribution of schools, classrooms, teachers, or other educational units in relation to student achievement (or other student outcomes). This is performed while also controlling for other (out-of-school) sources of growth in respect of student achievement, including prior student achievements, as well as student and family characteristics. The model produces estimates in respect of school productivity—i.e., the indicators of value added—under the counterfactual assumption that all schools serve the same group of students (Lissitz 2015). Evaluation involves developing, improving, and maintaining teaching skills and behaviors that lead to students achieving set outcomes and goals, while guiding teachers toward continuous improvement. Negative teacher evaluations may indicate that the teacher is not meeting the accepted standards of performance in regard to the teaching profession and, if not improved, may lead to dismissal or the non-renewal of their teaching contract (DeMitchell et al. 2012). The teacher possesses certain professional abilities and skills, which are externally manifested in their observable and evaluable pedagogical work; this is also evidenced in the results they achieve with their students. However, it is not only about what the teacher knows and what they objectively demonstrate in respect of teaching management. No less important is what they themselves think about their own professional abilities and skills, as well as how to evaluate themselves (Nikodémová 2016). According to Bandura (1978), the most desirable form of teacher behavior in respect of the teaching process is when the teacher's belief in their own abilities to perform a certain activity is in conjunction with the idea of the result, where both are equally high in both prerequisites. Such teachers are self-confident in themselves and also confident in their performance. Moreover, they show a high level of effort, persistence, and cognitive engagement in activities. Teachers who believe they have strong/high perceived competence will act, think, and feel differently than teachers who possess an idea of low self-efficacy, especially in relation to their future (which they shape realistically). Teachers who do not believe, for example, in their ability to inspire students' desire to learn dwell on negative classroom situations, but teachers with high perceived self-efficacy believe that students will be motivated to learn (Pintrich and Schunk 2001). In this context, concepts such as the teacher's self-efficacy, collective efficacy, and academic self-efficacy appear. The teacher's self-efficacy represents a hidden layer of the teacher's personality; in fact, it is part of the teacher's identity (Korthagen 2001). The analysis of several studies demonstrates that teachers with a high degree of this type of competence try harder, are more persistent, are not afraid to experiment, take healthy risks, face obstacles and—what is most important—possess the drive to (and also can) motivate students to learn (Nikodémová 2016). In the context of schools, collective efficacy can be understood as a motivational characteristic that results from the subjective perception of teachers regarding their overall level of teaching effectiveness. In this sense, collective efficacy describes how teachers perceive their collective ability to use their resources in order to solve complex or challenging situations, as well as to create and enrich a successful learning environment (Meyer et al. 2022). Based on their research, Kim and Ra (2022) state that students with a strong academic self-efficacy in their studies tend to have a higher level of career readiness. Furthermore, and at the same time, higher academic self-efficacy can support decision-making processes and enable students to set a career goal. This means that if an individual has a higher level of academic self-efficacy, then they are able to make better career decisions. Meyer (1997) notes that value-added indicators were developed in order to fairly compare performances between educational institutions. In most of these value-added indicators, student performance in respect of

tests or examinations was used to estimate differences in respect of the performances between educational institutions. These value-added indicators are examples of performance indicators due to the fact that they are based on test or exam scores. In general, the quality of performance measures—such as value-added indicators—depends on the provision of adequate samples, appropriate levels of reliability, good validity, and positive reactivity (Fitz-Gibbon and Tymms 2002). However, Fitz-Gibbon (1997) also mentions the negatives of such an assessment. For example, the National Value Added Project found that many principals lowered admission scores in order to render the added value to appear better. Further, the principals also looked to increase the number of students who could achieve good exam results with relatively little effort on their part. Additionally, test-score-based indicators possess limitations in respect of the differentiated education systems in which students in different programs take different tests or exams. This is an important drawback for the usefulness of performance indicators in regard to educational accountability due to the fact that, according to Timmermans et al. (2014), it is not possible to compare the value added in respect of entire schools. An alternative to using the results of tests or examinations in respect of estimating indicators of added value that could overcome these disadvantages is to use the educational positions of students. This can be considered as an indicator of efficiency. In such a case, an indicator of the added value of the school can be achieved by utilizing the grade repetition rate. In addition, however, it is important to emphasize the active involvement of the public as part of the quality assessment (Jankelová et al. 2017). This is also confirmed by Hamalová and Belajová (2011), as well as Urbaníková and Štubňová (2018), who note that when measuring the performance of any public services (such public services would include education), the most important factor is the resulting effect or the quality of the service, which can be measured via satisfaction. In other words, if a citizen is dissatisfied with the service, they perceive it as ineffective. Therefore, the aim of this research is to identify long-term satisfaction with the availability and quality of kindergartens and primary schools in the micro-region of Červený Kameň, which is understood to be an important determinant of the development of the local economy. A qualitative assessment (i.e., survey) among the parents of pupils in a selected region of Slovakia was conducted for this study. In order to increase the objectivity of the obtained results, the survey was conducted twice with an interval of 7 years (2014 vs. 2021). Respondents evaluated their satisfaction or dissatisfaction with the availability and quality of kindergartens and primary schools. Based on the obtained data, we tested the hypothesis on whether there is a significant statistical connection between the gender of the respondents and their satisfaction with the availability and quality of kindergartens and primary schools. At the same time, we also tested the hypothesis on whether there is a significant statistical difference in the assessment of the availability and quality of kindergartens and primary schools in 2014 and 2021. The reason for conducting this qualitative survey between parents and children was due to the fact that—according to Hakim (2020)—the community has an important role to play in respect of education, due to the fact that the community affects the progress of the quality of education. Fatchurrohman (2018) also confirms that community participation and the role of other stakeholders—i.e., students, teachers and principals, parents, and government—have a significant impact on the implementation of educational programs. The effort to quantify phenomena that represent the "quality indicators" of teaching is scientifically attractive (i.e., the temptation of so-called hard data), as well as politically and economically desirable (i.e., the need to better distinguish educational models worthy of support). However, such an approach also possesses obvious limitations. The problems with the evaluation approach in the further professional development of teachers, as well as the school as a whole, can be divided into three groups: (a) systemic; (b) methodical; and (c) personal (i.e., socio-psychological, moral, etc.). Quantitative assessments can be a tool to compare a limited selection of educational outcomes between schools (or countries), but they do not help the independent development of a specific school (Brestovanský 2019). Germany and German-speaking countries have been discussing this issue since the late 1980s. They emphasize curricular reform

(i.e., a fundamental change in concept) and a further development of their educational programs. Up until now, the quality and the nature of assessment in these countries have focused mainly on school teaching (Rýdl 2001). However, now there is a relatively high interest in the quality of education instead. In several countries (e.g., Austria, Hungary, and others) quality assessment is already legally enshrined in respect of education. Many schools have started experimenting with introducing models such as TQM, EFQM, and ISO 9000 from the industry to education in practice. The primary concern of every school should be to know the expectations, needs, and wishes of students as based on accurate and reliable data, not only on data that are based on the intuition and experience of the teachers. In order to improve matters in such a way requires regular feedback (i.e., investigating pupils' attitudes towards teaching and the school, as well as investigating pupils' ongoing development of knowledge, skills, and abilities) (Arendášová 2010; Orszaghova et al. 2017).

## 2. Theoretical Background

Although the term "education management" has been used in pedagogy only for a relatively short time, the education management approach—according to Katsaros (2008)—can be seen as a coordination of various factors: man, technology, and material. In addition, according to Isatayeva et al. (2018), education management is a set of principles, methods, organizational forms, and management methods that are aimed at increasing the effectiveness of teaching staff in their professional activities, but also in the development of personal qualities. Obdržálek and Horváthová (2004) perceive education management as a type of specific management that is oriented to the system of management and to the administration of education. It includes, in particular, the management of education and training, material and financial resources, as well as the provision of the legal framework for the purposes of education, personnel work, and also in respect of the management of people. Part of the overall school management is, therefore, the school management, which is focused on the internal management of the school. A successful provision of school activities consists of: compliance with legal regulations; ensuring the fulfillment of the education program; fulfilling the work duties of school staff; ensuring material and technical requirements; understanding personnel conditions; creating a better culture in respect of the institution; and developing good interpersonal relationships. The management of the school also includes the exercise of the competencies of the principal, their deputies, and other employees of the school. Makarenko (2020) notes that management in the education system at any level is characterized by two interacting sub-systems—i.e., management and control. Both are part of the structure of a holistic integrated system. This system requires targeted interaction of these two subsystems. In addition, Petrasová (2011) notes that teachers are responsible for bringing innovation to the educational process and, in this context, they are expected to constantly improve their profession (see also Bečica and Vavrek 2021). In democratic countries, two basic requirements are placed on education systems: improving the quality of education and its fair distribution to all according to their skills and abilities. Justice in education itself—or the otherwise effectiveness of the education system for all pupils—is increasingly considered a necessary condition for the quality of education. However, this is impossible without the further education of pedagogical staff.

A matter that is equally important in terms of increasing the quality and performance of schools is the perception of the professional competencies of teachers and school principals. In the period of 2015–2016, Stranovská et al. (2017) conducted a survey in regard to the perception of the importance of selected professional competencies within a sample of 730 teachers and 146 principals (both from primary and secondary schools). The results of their research showed that the perception of the degree of importance of professional competencies was different. Differences were found in the degree of perception in respect of the importance of the following competencies: ability to identify the psychological and social factors in respect of students' learning; ability to plan and implement their own professional development; and ability to use material resources in the teaching process.

According to Semarang (2012), teachers should always strive to increase their knowledge and skills in order to increase their ability to deliver quality education. Additionally, according to Pavlov (2012), the goal of the system in respect of the professional development of teachers is found in creating optimal conditions for the development of professional competencies as a key element in the development of the school system. In addition, it is also important in respect of increasing the quality of pedagogical work.

*2.1. Performance of Education*

Improving the quality of education leads to an improvement in the level of social and economic development of individuals and countries. Moreover, there have only been a few studies conducted in order to identify effective pedagogical and managerial practices that can guide learning actors in their activities (Povedano et al. 2021). The aim of education itself is to increase, acquire, deepen, and maintain the necessary knowledge, skills, and abilities (Bartoňková 2010) to function in today's society. Performance in the learning process is quantified by classification levels or a weighted study average (Starecek et al. 2019). Based on the structure of the student's value orientation, the teacher can motivate and stimulate them and thus increase their performance (Vnoučková et al. 2017).

When evaluating education, it is necessary to understand that quality is the most important variable that prevails over quantity (Salgado and Novi 2015). The concept of quality in the field of education addresses the structures, processes, and results of education (Davok 2007). In this sense, two different dimensions are considered: (a) from an institutional point of view, in terms of the quality of education, and (b) in terms of student generated results. The first dimension considers the efficiency, effectiveness, efficiency, and relevance of educational systems and institutions; whereas, the second dimension is in regard to the development of learning skills, the knowledge of scientific and literary culture, and the technical knowledge required for the labor market, as well as that which is required in respect of social transformation (Povedano et al. 2021). Darling-Hammond and Ascher (1991), as well as Dourado et al. (2007), advise that the factors influencing the performance of education include: the social aspect, income levels, social inequality, the school management process, the curriculum, and the preparation of teachers for the purposes of teaching. Another relevant factor in regard to increasing educational performance is helping parents increase their child's interest in education. According to Spencer et al. (2011), as well as Van Voorhis et al. (2013), good mentoring from an early age can increase children's motivation to learn and, at the same time, parents can better evaluate the quality of teaching materials. The infrastructure built in schools can also contribute to increasing the efficiency of education, especially in the long run, as it brings with it opportunities for innovation in education (Milawati and Fahrudin 2021). According to Salgado and Novi (2015), the identification of these factors is crucial, as it can contribute to increasing the quality and performance of the educational process in the context of improving student outcomes, thus creating a systemic view of the field of education and thus measuring its quality. The evaluation of the quality of education remains a controversial issue in the literature. This is due to the fact that although several extensive studies have been conducted, they generally only evaluated the performance of specific subjects, such as math and reading (Fernandes and Gremaud 2009).

*2.2. Regional Education in the Conditions of Slovakia*

Until 1989, all educational institutions in Slovakia were centrally managed. According to Obdržálek (2014), the basic shortcomings of school management in this period included the absence of real, not simply postulated, democracy—that manifested in the participation of teachers in management, especially in the processes of decision making in education. Furthermore, Obdržálek also included the inefficiency of work with information, which was manifested in its irrationality, the lengthiness and bias, and the low effectiveness of control activities, which were carried out according to formal indicators (i.e., the absence of objective criteria for the selection of senior education staff, especially school principals, and

the absence of goal orientation of management at the various stages and levels). This was exacerbated by the cumbersomeness of the three-tier system of national committees who were involved in school management, as well as the unilateral emphasis on tasks planned by the higher authorities instead of an emphasis on tasks arising from the needs of the school and its other facilities.

Slovakia's transition from a command economy to a market economy changed the nature of the economy and thus required changes in the management of public administration. The central model of public administration that was applied in Slovakia until 1989 was typical in its centralism and bureaucracy, which essentially copied the then management of the country's economy. Following this change in the economy, emphasis began to be placed on the endogenous development of the territory, which was based on the use of local resources within the framework of the sustainable development of the territory. Until 1989, education in Slovakia was perceived, mainly from an institutional point of view, as a deliberate activity of the educator. This meant that it privileged the role of society, the teacher, and other external influences on the child through the content of education (Kosová and Kasáčová 2007). However, the decentralization that took place on 01/07/2002 in respect of education management also brought changes in this area. Within this process, the competencies and responsibilities in certain areas of education were transferred from the state administration to local and regional self-government. Schools and educational establishments were transferred to the competencies of municipalities and regional authorities, which thus became the founders of the majority of state schools and educational establishments. Since 1st January 2004, a new Act (Act No. 596/2003 Coll.) regarding state administration in education and school self-government, as well as a decree (Decree No. 291/2004 of the Ministry of Education, Science, Research and Sport of Slovakia) were brought into law. These legal acts determined the details in respect of the method of establishing school self-government bodies, as well as their composition, organizational, and financial support. At the same time, the founding powers of the state administration bodies were also changed. In addition, the founding authority of schools instead became regional school offices, municipalities, state-recognized churches or religious societies, and/or a natural or legal person. This law also established regional school authorities as bodies of specialized state administrations (Žáčková et al. 2009). Based on Act No. 245/2008 Coll. in respect of education and training (i.e., the School Act), the system of schools in Slovakia was defined, which also included the successive levels of education. According to Lauko et al. (2010), the development of the school network (i.e., the number of schools, classes, and teachers), reflects not only changes in the number of pupils, but also spatial specifics in the national and ethnic structure. This is in addition to the age structure of the population, and other factors, such as the finances of the local government and its priorities in the field of education, transport infrastructure, the condition of school buildings, etc. Tomková (2019) adds that the school reforms implemented in Slovakia between 1997 and 2008 brought about major curricular changes in education. These changes were implemented to promote science and technology education in all types of schools and thus to promote the science and technology education of pupils. However, the practice has shown that the changes introduced in the education system have not produced the expected results; furthermore, pupils' lack of interest in technical subjects persists. According to Váňová (2006), almost all countries are gradually undergoing some degree of a decentralization of education, thereby aiming to create and preserve a wide scope of pedagogical, administrative, and financial autonomy. Thus, in most cases, central authorities determine the general arrangement of the school system and set the main lines of school policy and planning. Then, regional and local authorities put these general directives into practice and elaborate on them. Finally, schools—now endowed with a considerable degree of autonomy—thus work out their own educational projects. According to Naper (2010), an interest in decentralization in regard to education is also growing in Norway, where schools are taking steps to increase their autonomy and reduce central regulation, thereby hoping that this will result in better school management and greater educational efficiency.

*2.3. Pre-School and School Facilities at the Level of Local Self-Governments within Slovakia*

In respect of the conditions of Slovakia, the municipality primarily fulfils the role of the founder in the field of education, which it performs within the framework of the exercise of the original competencies (i.e., kindergartens, primary art schools, language schools at primary schools, and school facilities), as well as within the framework of the transfer competencies of state administrations (i.e., primary schools). At present, certain local authorities are closing their school facilities due to low interest among children (these are mainly small municipalities far from growth poles whose population is primarily declining because of poor socio-economic conditions). Still, on the other hand, there are also municipalities close to growth poles, or so-called "satellite municipalities", which have insufficient space to allow all their children to attend school directly in the municipality. Balážová and Lazárová (2014) add that many of the problems that have been caused are mainly due to a lack of knowledge of the relevant laws governing the operation and specifics of educational institutions, their operation, financing, and staffing, as well as the matters related to their material and technical support.

Municipalities in Slovakia may operate kindergartens as separate legal entities or, if there are more of them in a municipality, all kindergartens may be merged with the consent of the founder; when this is completed, one legal entity with detached workplaces will have been created. Some municipalities merge kindergartens with primary schools into a single legal entity of a primary school with a kindergarten. Kindergartens without a legal entity are usually part of the municipal office. Kindergartens provide pre-primary education based on the national curriculum for pre-primary education in kindergartens, which is detailed in a binding document in respect of the development of school curricula. Its main objective is to achieve an optimal cognitive, sensorimotor, and socio-emotional level as a basis for primary school and life in society. According to Guziová (2011), the school educational program has specifically defined frameworks in terms of structure—which is provided based on the Education Act in the State Educational Programme ISCED 0, Pre-primary Education. However, in respect of the content, didactic, and professional methodological points of view, it can be conceived in a very different way. According to Muchacka (2013), public kindergartens should be universal institutions. Furthermore, local self-governments should be obligated to run kindergartens, but on the condition of obtaining sufficient financial resources. The strategy document "Strategy 2020" details, in one of its guidelines, that in the school year 2019/2020 at least 95% of all four-year-old children in EU countries will be subject to compulsory pre-school education.

The development of the number of kindergartens in the period of 1989–2019 showed a fluctuating trend (Figure 1). From 1989 to 1995, the number of kindergartens slightly decreased each year. However, a change occurred in 1996 when the number of kindergartens increased by ten compared to the previous year. However, between 1997 and 2008, there is again a slight annual decline in the number of kindergartens. In 2009, the number of kindergartens increased by two. The total number of kindergartens is mainly related to the demographic development in society and the economic conditions of the founders. When the interest in kindergartens declined, municipalities attempted to solve this situation by merging kindergartens with primary schools or, when the number of kindergartens increased, they were closed down. This trend was also evident in 2010–2012 when the number of kindergartens decreased every year. However, the decline in the number of such facilities is now beginning to negatively impact the country as the number of children who cannot be accommodated primarily in kindergartens has recently been increasing. Through the Ministry of Education, Science, Research, and Sport, the state has already responded to this situation by allowing municipalities, through subsidies, to reconstruct or build new kindergartens in order to meet parental demand. Hudakova (2016) also confirms that the lack of places in kindergartens is a consequence of the demographic development of the population in the 1990s, which is when many facilities were liquidated. Today, local self-governments are struggling with the shortage as the birth rate has increased. The demographic development of the reference group of 3–5-year-olds is currently in the growth

phase. This phase places increased demands on the number of kindergartens, as well as increased demands on financial resources. Regarding kindergarten development forecasts, the number of kindergartens is projected to increase between 2020 and 2021, peaking in 2022 at a number of 3256 kindergartens. After that, the number of kindergartens is projected to decline in the following years (Herich 2019). Additionally, when comparing the periods of 1989 and 2019, there was an overall decrease of 25% in kindergartens. In addition to municipalities, private persons or churches can also be founders of kindergartens under certain conditions. Until 1989, all kindergartens were state run. After the restoration of local self-government in 1990, private and church-run kindergartens began to appear. In 1993, the first private kindergarten appeared in the records and in 1995, the first two church kindergartens. The number of private and religious kindergartens has been changing dynamically, but in 1989–2019 the share of state-run kindergartens was at an average level of 97.19% (Figure 1). While kindergartens belong to the original competencies of the local self-government, primary schools and municipalities are now involved in the process of decentralization of public administration. It must be noted that the compulsory school attendance in Slovakia is ten years. For example, in the Netherlands, compulsory schooling lasts at least twelve years and starts when a child reaches the age of five, although most children attend school from four. Primary education thus lasts eight years for children aged four to twelve. By contrast, Denmark has nine years of compulsory schooling for children aged seven to sixteen. This education takes place in primary and lower secondary school (Folkeshole), which includes pre-school classes and classes in grades 1 to 7, or it can include pre-school classes and classes in grades 1 to 10. The completion of the ninth grade represents a completion of compulsory schooling; in other words, 10th grade is optional. In the former case, after completing grade seven, pupils have to change schools and complete their compulsory education (Kožuchová and Pavelka 2007).

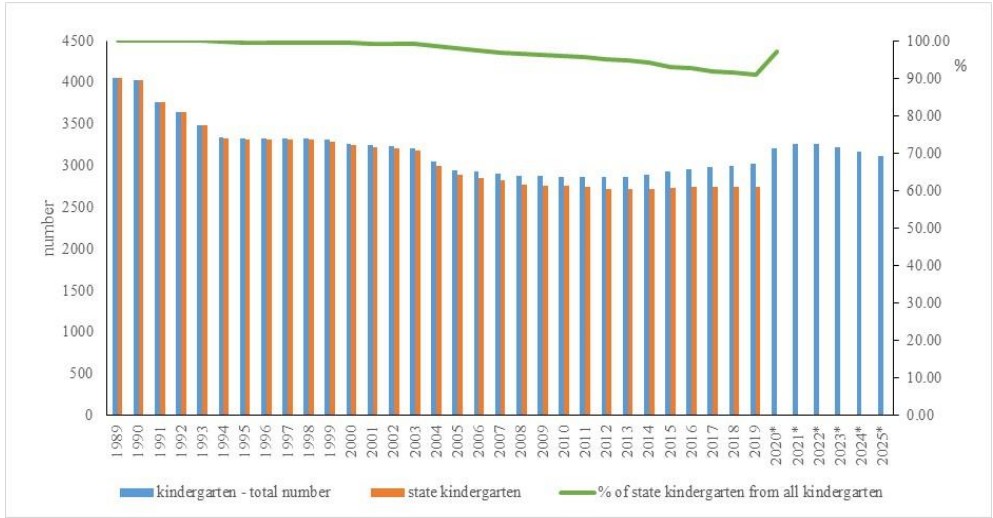

**Figure 1.** Development of the number of kindergartens in Slovakia during the period of 1989–2025. Legend: * it's a forecast, Sources: Slovak Centre of Scientific and Technical Information (SCSTI) (2021); Herich (2019); and own processing.

Primary schools, in the conditions of Slovakia, provide basic education in accordance with the principles and objectives of education and training according to Act No. 245/2008, Coll. They support the development of the pupil's personality based on the principles of humanism, equal treatment, tolerance, democracy, and patriotism, which are delivered in terms of reason, morality, ethics, aesthetics, work, and physical activities (Bernátová 2013). Primary schools in Slovakia possess two levels: level 1 (primary education) and level 2 (lower secondary education). Education in both grades is carried out according to separate state educational programmers, which build on each other. Makišová (2013) notes that the basis of the primary education program is to ensure a smooth transition from pre-school

education and family care to school education by stimulating children's cognitive curiosity based on their knowledge and their own experiences. The program's base is the progression from the known to the unknown in terms of acquiring experience from observing and experiencing activities, as well as events linked to the child's life and their immediate cultural and natural environment. By transferring competencies in the field of primary schools to the local self-government level, Slovakia has become one of the developed European countries. Examples include Germany and Austria, where municipalities are the founders of schools, but regional and central authorities have retained certain powers (Bartušová 2017). There are nine years of compulsory schooling in Austria, with the first stage (one–four years) at the Volksschule (folk school), which provides elementary education. Pupils who are between five and eight years old can be completed either at the second stage of the Volksschule or at the second stage of primary school. Pupils can also continue to a general education upper school, where they complete a lower level (Kožuchová and Pavelka 2007). The municipalities fully manage Swedish schools, which set the schools' goals, finances, and also evaluate them (Bartušová 2017). Compulsory schooling in Sweden is implemented in a nine-year primary school (grundskola) from the ages of 7 to 16 (as an aside, from 1998 onwards the starting age changed from age seven to the age of six). Primary school is not divided into different grades, but pupils' school performances are evaluated after grades 5 and 9, respectively (Kožuchová and Pavelka 2007). In Finland, municipalities also manage schools but are partly financed by the government (Bartušová 2017). In Finland, primary schools (Peruskoulu/Grundskola) are divided into two basic levels—lower level (grades 1–6) and upper level (grades 7–9)—with municipalities and schools possessing considerable authority in setting the curriculum (Kožuchová and Pavelka 2007).

When compared to kindergartens, we observe a different development within primary schools. While the number of kindergartens declined each year between 1989 and 1995, the number of primary schools increased. According to Lauko et al. (2010), the initial increase in the number of primary schools was related to the rise in the number of pupils and the establishment of religious and private schools, which was made possible after 1989. The exception was 1994, when a slight decline in the number of primary schools was recorded. However, after a renewed increase in the number of primary schools in 1995, 1996, and 1998, we observed a slight decrease in the number of primary schools each year between 1999 and 2019. Thus, there was a 9.9% decrease in the number of primary schools between 1989 and 2019. Similar to kindergartens, primary schools can also be arranged by private persons or the church under certain conditions. The first private primary school appeared in the statistics in 1993. In addition, the first two church primary schools were recorded even as early as 1990. The number of private and religious primary schools has been changing dynamically, but in 1989–2019 the share of state primary schools was at an average level of 94.79% (Figure 2). The projection of the number of primary schools assumes that the annual decline that has been recorded over the last 20 years will stop, which would thus entail an increase in the number of primary schools. While in 2020, it is assumed that there will be 2166 primary schools in Slovakia; in 2022, however, there should be 2184. However, the forecast foresees a slight decrease in the number of primary schools in 2023, but an upward trend in the following two years (Figure 2).

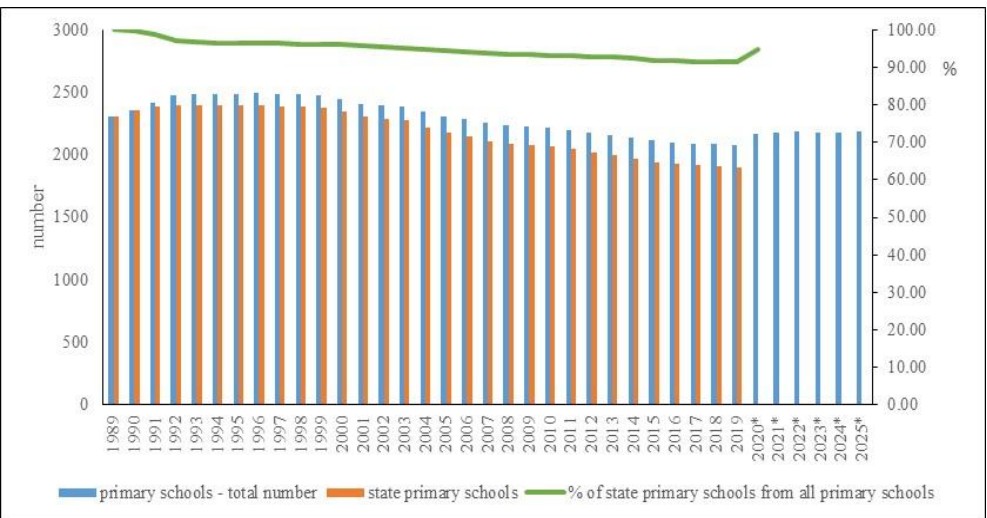

**Figure 2.** Development of the number of primary schools in Slovakia in 1989–2025. Legend: * it's a forecast, Sources: Slovak Centre of Scientific and Technical Information (SCSTI) (2021); Herich (2019); and own processing.

### 3. Methodology

The present research aim is to identify long-term satisfaction with the availability and quality of kindergartens and primary schools in the Červený Kameň microregion, understood in the context as an important determinant of local self-government performance. This study was conducted using a qualitative evaluation (survey) among the parents of the students in a selected region of Slovakia. In order to increase the objectivity of the obtained results, the survey was conducted two times with an interval of 7 years (2014 vs. 2021). The dependent variable measured in this case is the respondent's satisfaction in the form of a nominal dichotomous response (0/1). This is interpreted in time and depends on the independent variable or identifying factor, which is the respondent's gender (i.e., male/female).

The questionnaire survey, together with the observation and experiments, represents a comprehensive research method for the collection of primary data, which is often used in the field of education (Jansen et al. 2017; Huisman et al. 2019; Janštová and Šorgo 2019; Linberg et al. 2017; Tomšik 2017). When conducting the two independent questionnaire surveys sufficiently far apart, it subsequently allows for another scientific method, i.e., comparison (Kok et al. 2013; Bofferding and Farmer 2018; Crompvoets et al. 2020; Wolff 2021). By comparing the obtained results, it is possible to identify progress in satisfaction with the selected attributes (i.e., in the form of a measured dependent variable) and thus offer solutions corresponding to the real situation in the field. An example of this is found in the research of authors such as Quesada Serra et al. (2013), Vispoel et al. (2018), Miller-Young et al. (2021), and many others.

The questionnaire survey was conducted online in 2014 and 2021 through the Google-Docs platform, which allows reaching the survey sample free of charge and without restrictions. The questionnaire itself was placed for two months on the website of the Červený Kláštor microregion, as well as on the websites of its member municipalities (which represents a total of nine municipalities—Báhoň, Budmerice, Častá, Doľany, Dubová, Jablonec, Píla, Štefanová, and Vištuk). A total of 993 respondents, citizens of the municipalities in the abovementioned microregion, participated in the questionnaire survey, with 368 respondents taking part in the survey in 2014 and 625 in 2021. The questionnaire survey was comprehensive and focused on assessing the place of housing, basic living needs, work, or social space in the different areas. For the present research, we focused on the section where respondents were asked to rate their satisfaction using a choice of two options (satisfied/dissatisfied):

(a)    with the availability and quality of kindergartens;
(b)    with the availability and quality of primary schools.

Due to the nature of the collected data, Cramer's V coefficient was used to evaluate the obtained results, which, according to Tomšik (2017), is a non-parametric test that detects the degree of association between two nominal variables and is based on Pearson's Chi-square statistic. In order to assess differences over time (specifically in regard to the results of the questionnaire surveys that were conducted in 2014 and 2021), the Levene test and selected moment characteristics were used (Maros et al. 2021). All analyses were processed in MS Excel, Statistica 13.4 and Statgraphics XVIII.

## 4. Results

Over two months in 2014 and 2021, residents of the Červený Kláštor microregion were asked to rate several aspects of life in the microregion, with a basic description of the results shown in Table 2.

**Table 2.** Basic description of the Červený Kláštor micro-region.

|  | 2014 | | 2021 | |
| --- | --- | --- | --- | --- |
|  | **Pre-School Facilities** | **Primary Schools** | **Pre-School Facilities** | **Primary Schools** |
| Count | 325 | 315 | 564 | 549 |
| Average | 0.84 | 0.80 | 0.90 | 0.86 |
| Mode | 1.0 | 1.0 | 1.0 | 1.0 |
| Std. deviation | 0.35 | 0.39 | 0.29 | 0.34 |
| CoV (%) | 42.20 | 48.58 | 32.23 | 40.12 |
| Minimum | 0 | 0 | 0 | 0 |
| Maximum | 1.0 | 1.0 | 1.0 | 1.0 |
| Range | 1.0 | 1.0 | 1.0 | 1.0 |
| Skewness | −1.96 | −1.58 | −2.79 | −2.09 |
| Kurtosis | 1.85 | 0.51 | 5.80 | 2.41 |

Source: Own processing.

Each respondent was given the opportunity to voluntarily answer each question, as documented by the different numbers of respondents answering each question (see line "Count" in Table 2). Provided the fact that we were interested in the difference in ratings of the availability and quality of pre-schools and primary schools, the hypothesis tested was:

**Hypothesis 1 (H0):** *There is no significant statistical association between the respondent's gender and their satisfaction with the availability and quality of kindergartens and primary schools.*

**Hypothesis 1 (H1):** *There is a significant statistical association between the respondent's gender and their satisfaction with the availability and quality of kindergartens and primary schools.*

The use of Cramer's V coefficient does not require testing the normal distribution of the individual in regard to dependent or independent variables. It is based on the absolute frequencies in the groups that are illustrated in Table 3, which are then used to calculate the test characteristic (i.e., the Chi-square).

Based on the frequencies illustrated in Table 3, a test characteristic (Chi-squared) is calculated. The association rate (expressed by the Cramer V coefficient) can be trivial in each case. Satisfaction with pre-schools or primary schools is not determined by the gender of the respondent, i.e., the parent of the pupil attending the facilities in question. We observe similar values of the Cramer's V coefficient in both evaluated years; i.e., the situation that is being investigated persists. Parents perceive, in the long term, the accessibility and quality in the same way (meaning no statistically significant differences were confirmed); i.e., research hypothesis H0 cannot be rejected.

**Table 3.** The absolute abundances for the evaluation of the stated hypothesis in 2014 and 2021.

| | | **2014** | | | **2021** | |
|---|---|---|---|---|---|---|
| | | **Dissatisfied** | **Satisfied** | | **Dissatisfied** | **Satisfied** |
| Pre-school facilities | men | 16 | 120 | men | 22 | 221 |
| | women | 33 | 156 | women | 31 | 290 |
| | | Dissatisfied | Satisfied | | Dissatisfied | Satisfied |
| Primary schools | men | 23 | 111 | men | 29 | 208 |
| | women | 37 | 144 | women | 47 | 265 |

Source: Own processing.

In a second step, we looked at changes over time, which can already be partially evaluated using the data in Table 3. Due to the nature of the data—i.e., a nominal dichotomous variable—it is not possible, in our opinion, to subject them to tests in respect of the comparison of means (i.e., the *t*-test and Mann–Whitney test). In addition to the mean values and selected moment characteristics, homoskedasticity can also be observed using the Levene test and through the following hypotheses:

**Hypothesis 2 (H0):** *There is no significant statistical difference in the variance of the questionnaire results (meaning the evaluation of availability and quality of kindergartens/primary schools) in 2014 and 2021, i.e., $\sigma^2_{2014} = \sigma^2_{2021}$.*

**Hypothesis 2 (H1):** *There is a significant statistical difference in the variance of the questionnaire results (meaning the evaluation of availability and quality of kindergartens/primary schools) in 2014 and 2021, i.e., $\sigma^2_{2014} \neq \sigma^2_{2021}$.*

We assume a lower dispersion in the case of a more explicit assessment of the availability and quality of kindergartens and primary schools. As such, rejecting the null hypothesis de facto also implicitly constitutes a confirmation of the differences in the evaluation that separately focuses on pre-schools and primary schools (see Figures 3 and 4).

The baseline moment characteristics and the relative abundances (Figure 3) indicate differences in the 2014 and 2021 survey results. The skew of the 2021 results is more pronounced, thereby showing a more significant number of above-average ratings, i.e., more satisfied respondents/parents. This is accompanied by lower variability and thus a rejected homoskedasticity (LE = 25.904; $p < 0.01$); i.e., the differences are statistically significant.

We observe a similar level of moment characteristics in the case of satisfaction with the availability and quality of primary schools, which is illustrated in Figure 4. The results for the 2021 assessment are more negatively skewed and are thus also characterized by lower variability. These characteristics caused the confirmation of differences in the variance of the ratings (LE = 15.927 and $p < 0.01$), which can also be seen in the better average satisfaction of the respondents. Based on these results, hypothesis (H0) can be rejected; i.e., satisfaction with both pre-schools and primary schools is different and varies over time. The trend of change is positive and hence desirable.

The results of the aforementioned analyses are also confirmed by the development of the number of kindergartens and primary schools in the microregion, as well as the number of children attending these facilities. In 2014, there were nine kindergartens and six elementary schools operating in the territory of the microregion, which were attended by a total of 1283 children (439 in kindergarten, 844 in elementary school). Parents' satisfaction with the availability and quality of these facilities was reflected in 2021. Despite the fact that the number of kindergartens and primary schools increased by only one facility, there was a significant increase in the number of children. A total of 530 children attended kindergartens in 2021, which represents an increase compared to 2014 by 20.72%. A total of 1267 children attended primary schools, which is an increase compared to 2014 by 50.12% (Figure 5).

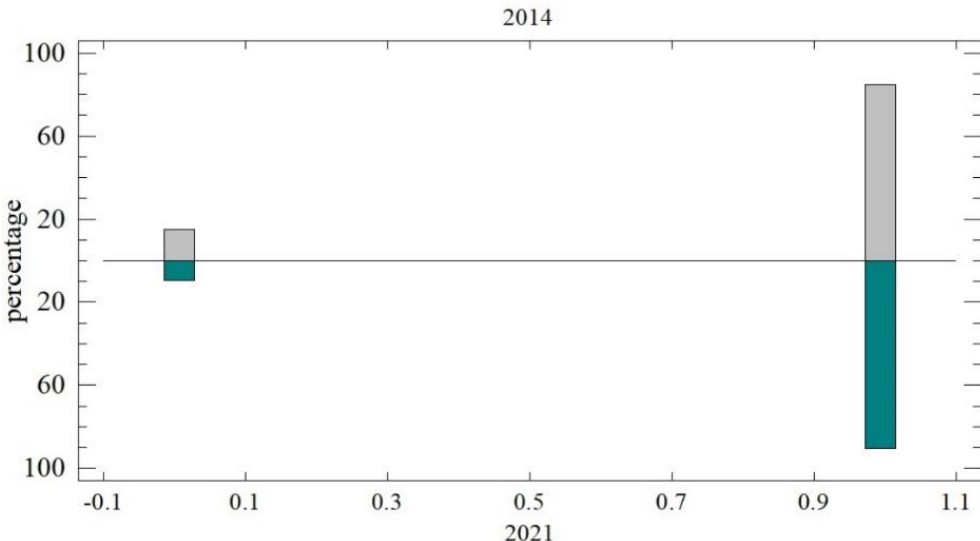

**Figure 3.** Comparison of the satisfaction with pre-school facilities in 2014 and 2021. Source: Own processing.

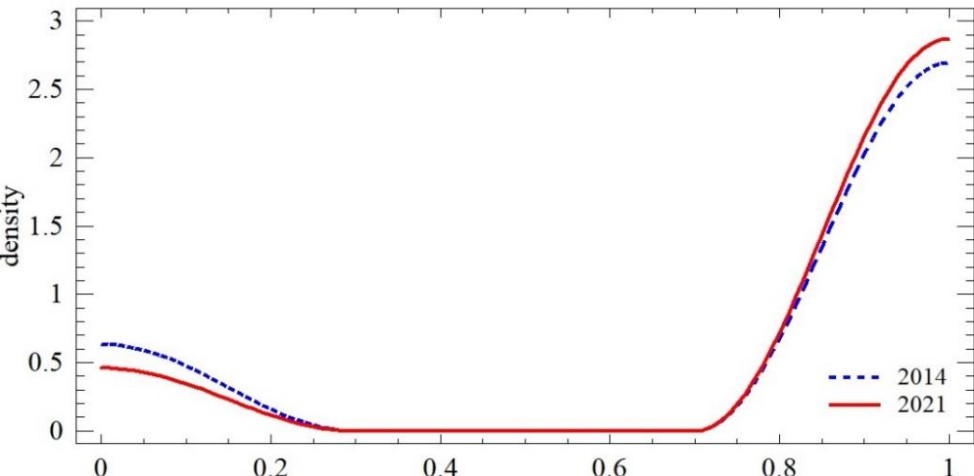

**Figure 4.** Comparison of the satisfaction with primary schools in 2014 and 2021. Source: Own processing.

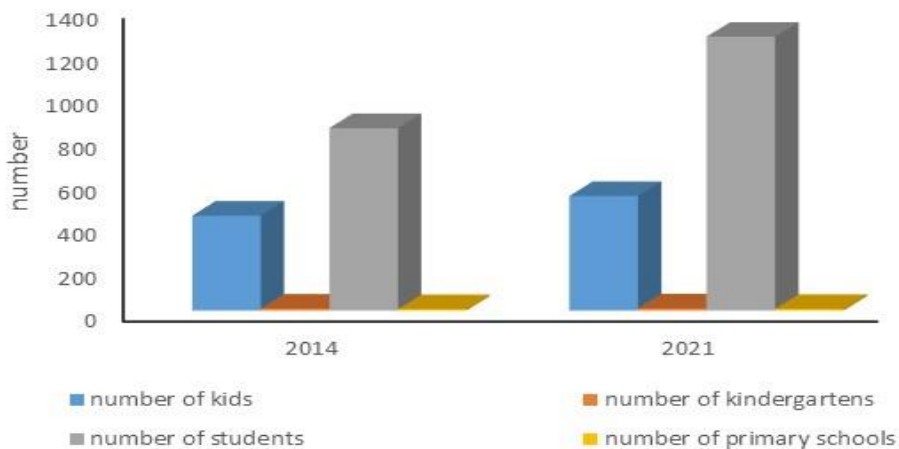

**Figure 5.** The number of children in kindergartens and primary schools in the microregion of Červený Kameň. Sources: Statistical Office of the Slovak Republic and own processing.

From the point of view of a territorial and administrative division, the municipalities of the microregion belong to the Pezinok district and the Bratislava self-governing region. The good transport accessibility of the municipalities and relatively few opportunities to obtain a job directly in the municipalities creates the conditions for a high number of residents leaving for employment outside the municipalities. This is also confirmed by the trend of workers leaving for work in 2011 (this figure is determined only during the population census), when 87.1% of workers left the microregion for work. These are primarily younger age groups that have families and have the opportunity to drive their children to school facilities in the surrounding towns. Nevertheless, based on the development of the number of children attending kindergartens and primary schools, it can be concluded that the majority of children living in the municipalities attend local schools. Additionally, one of the reasons for this may be the results of a certain qualitative survey, which confirmed that, when compared to 2014 and 2021, parents perceive a higher quality of education in the kindergartens and elementary schools in the microregion. This finding is also confirmed by Lynch (2015), who states that in terms of quality evaluation, smaller schools have better student success rates. Students in smaller schools, and thus in smaller groups, have better relationships with their classmates and teachers. This may also be one of the reasons why the number of school facilities and the number of children attending them increased in 2021. Based on the facts found, the subject of future research in this area will be the identification of parents' reasons for placing their child in a given school.

Economic aspects within the framework of regional education are closely connected with the effective management of schools and have an impact on the socio-economic development of the territory. Kindergartens, elementary art schools, language schools and school facilities under the jurisdiction of municipalities since 1 January 2005 are financed from share taxes. Funds are provided to the founders of primary schools from the state budget. These financial resources are provided by the state in the form of a state subsidy for the founders, who subsequently redistribute them to primary schools. The founder is obliged to distribute all normative financial resources among the schools of which he is the founder. These financial resources cannot be used to finance kindergartens and school facilities, as these schools and school facilities are financed from municipal tax revenue. Additionally, in this context, the existence of pre-school and school facilities at the local level becomes an important part of the socio-economic development of the territory. As long as there are pre-school and school facilities in the municipality, the number of inhabitants of the municipality with permanent residence also increases. This is an important aspect for municipal financing. Under the conditions of Slovakia, municipalities are financed from the state budget based on the number of inhabitants with permanent residence. The more inhabitants, the more funds municipalities receive from the state budget in the form of a share tax. From the total amount of funds that the state collects in the form of personal income tax, it redistributes 70% to the level of municipalities based on the number of people with permanent residence in the municipality. These revenues make up 80% of the total revenues of the municipalities, and since they are not earmarked, the municipalities use them precisely for their socio-economic development.

## 5. Conclusions

The evaluation of availability and the quality of school facilities in the context of the self-government level is a very complicated process. From an economic point of view, the performance of school facilities determines the number of students for whom schools receive funding, but this does not indicate the quality of education. It is the quality of education that can be evaluated by qualitative methods, where we involve parents or students in the evaluation process; as such, this was also the focus of our research. By utilizing a questionnaire survey, we identified satisfaction with the availability and quality of pre-schools and primary schools in the microregion of Červený Kameň from the point of view of the students' parents. The data obtained show that parents perceive, in the long term, accessibility and quality in the same way (with no confirmed statistically significant

differences). At the same time, we find that satisfaction with both kindergarten and primary schools varies and changes over time. These results need to be understood in the context of the size of the survey population and the structure of the questionnaire itself. In the future—in order to increase the meaningfulness of the achieved results—we plan to focus more on other determinants in respect of the performance of school facilities at the local self-government level. One of them is, for example, the number of Ukrainian children that are now placed in school facilities and their adaptation within the Slovak education system. In the future, this issue will be a very important determinant in respect of the development of education at the local level. This is relevant due to the fact that during the course of the military conflict, 51,801 Ukrainians have, as of yet, requested temporary refuge in Slovakia, of which 22,525 were children.

**Author Contributions:** Conceptualization, V.P., M.D. and R.V.; methodology, R.V.; software, R.V.; validation, R.V.; formal analysis, R.V.; investigation, V.P., M.D., R.V., J.M., P.H., M.V. and T.V.; resources, V.P., M.D., R.V., J.M., P.H., M.V. and T.V.; data curation, R.V.; writing—original draft preparation, V.P., M.D. and R.V.; writing—review and editing, V.P., R.V. and M.D.; visualization, R.V.; supervision, R.V., V.P. and M.D.; project administration, V.P. and R.V.; funding acquisition, R.V. All authors have read and agreed to the published version of the manuscript.

**Funding:** This research was funded by VSB—Technical University of Ostrava, grant number SP 2022/29; Ministry of Education, Science, Research and Sport of the Slovak Republic, grant number VEGA 1/0517/22.

**Institutional Review Board Statement:** Not applicable.

**Informed Consent Statement:** Not applicable.

**Data Availability Statement:** Data is unavailable due to privacy.

**Conflicts of Interest:** The authors declare no conflict of interest.

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
