# Peer review of "Availability and Quality of School Facilities as a Determinant of Local Economic Development: The Slovak Experience"

_economies, doi:10.3390/economies11020035_

Round 1

Reviewer 1 Report

In my view, the contents of the paper’s sections are not sufficiently aligned between each other. In this sense, it turns out difficult to follow the logic behind the text. One example is the following sentence in line 30, which seems unclear to me and disconnected from the rest of the text: In Finland, only 10% of the best graduates provide basic education. 

Furthermore, in section 2, authors start to talk about Economic growth, which also seems to be disconnected from the context. In my view, economic growth and other important variables should be treated in strong relation to the main hypothesis, where emphasis should be placed on the posed hypothesis. That is, the discussion should possibly motivate the analysis and raise more interest in the topic.

Moreover, it takes a long reading until the reader get a slight clue on paper’s main hypothesis. It appears in page 10 (section 3). I believe that the authors should highlight their hypothesis long beforehand, possibly in the Introduction, to connect thus the earlier text and affirmations with such hypothesis. This would clarify the reason why this study is interesting, and it would strengthen the motivation why this analysis was undertaken.

Finally, in my view, the authors should rewrite the paper to address the previous issues and at the same improve typos and grammatical incoherencies along the text.

Author Response

Dear reviewer, 

thank you very much for your valuable comments. Each of them has been incorporated, namely: 

C1: In my view, the contents of the paper’s sections are not sufficiently aligned between each other. In this sense, it turns out difficult to follow the logic behind the text. One example is the following sentence in line 30, which seems unclear to me and disconnected from the rest of the text: In Finland, only 10% of the best graduates provide basic education. 

R1: The entire introduction has been corrected and supplemented/extended.

C2: Furthermore, in section 2, authors start to talk about Economic growth, which also seems to be disconnected from the context. In my view, economic growth and other important variables should be treated in strong relation to the main hypothesis, where emphasis should be placed on the posed hypothesis. That is, the discussion should possibly motivate the analysis and raise more interest in the topic.

R2: The mentioned section was removed and the parts relevant in relation to the solved issue were added, see lines 50-123.

C3: Moreover, it takes a long reading until the reader get a slight clue on paper’s main hypothesis. It appears in page 10 (section 3). I believe that the authors should highlight their hypothesis long beforehand, possibly in the Introduction, to connect thus the earlier text and affirmations with such hypothesis. This would clarify the reason why this study is interesting, and it would strengthen the motivation why this analysis was undertaken.

R3: The hypothesis was supplemented and justified in the introduction, see lines 200-214.

C4: Finally, in my view, the authors should rewrite the paper to address the previous issues and at the same improve typos and grammatical incoherencies along the text.

R4: The manuscript has been checked by a native speaker.

Reviewer 2 Report

-        Null an Alternative hypothesis must be corrected. 

-        The results include only one variable with two categories to evaluate the satisfaction of the facilities. The analysis has to be more comprehensive; collecting more data would be ideal.

Author Response

Dear reviewer, 

thank you very much for your valuable comments. We consider your comments as a great possible extension of our research. We will continue in this way in future research.

Reviewer 3 Report

Satisfaction with education does not have to be measured solely by the level of satisfaction with the service provided. An external (independent) evaluation system may also be implemented, allowing for the determination of the educational added value (EAV). The EAV measure is widely known in Great Britain and the USA. I believe that the considerations in this article should be supplemented with this issue.

Numerous scientific reports indicate that a more important element than equipping the school with material resources is properly motivated teaching staff and the students themselves. So I suggest supplementing the literature review with these issues as well.

I disagree with the statement that quantitative measurement methods only allow comparisons between education systems and schools. Exactly, that they perfectly allow to monitor quality and take improvement actions on an ongoing basis, provided that the results are frequently monitored and implemented improvement actions both in the teacher's work and students' involvement, as well as implementing a system to compensate for educational shortcomings among students, which requires additional expenditure. They suggest an interest in the American models of improving the quality of education in the context of educational added value.

I do not understand why the indicator of competences and the introduction of formal education of the surveyed population, but only the percentage of people holding a driving license and knowledge of foreign languages among young people. This issue should definitely be clarified in the article.

Author Response

Dear reviewer, 

thank you very much for your valuable comments. Each of them has been incorporated, namely: 

C1: Satisfaction with education does not have to be measured solely by the level of satisfaction with the service provided. An external (independent) evaluation system may also be implemented, allowing for the determination of the educational added value (EAV). The EAV measure is widely known in Great Britain and the USA. I believe that the considerations in this article should be supplemented with this issue.

R1: Added paragraph, see lines 124-136. 

C2: Numerous scientific reports indicate that a more important element than equipping the school with material resources is properly motivated teaching staff and the students themselves. So I suggest supplementing the literature review with these issues as well.

R2: Added paragraph, see lines 124-136. 

C3: I disagree with the statement that quantitative measurement methods only allow comparisons between education systems and schools. Exactly, that they perfectly allow to monitor quality and take improvement actions on an ongoing basis, provided that the results are frequently monitored and implemented improvement actions both in the teacher's work and students' involvement, as well as implementing a system to compensate for educational shortcomings among students, which requires additional expenditure. They suggest an interest in the American models of improving the quality of education in the context of educational added value.

R3: Added paragraph, see lines 124-136. 

Reviewer 4 Report

The manuscript makes a valiant attempt at uncovering “whether the availability and quality of school facilities is a determinant of local development...” Unfortunately, the manuscript appears to have virtually no substance, relying on two simple questions and the most basic of statistical analyses based on a survey of the residents of a small region seven years apart. The crucial question of a link – any link – of primary education to local economic development is never even touched. The authors should review the entire article for logical consistency and substantive findings, then consider submission again.

Author Response

Dear reviewer, 

thank you very much for your valuable comments. The article was completely revised and the parts related to the solved problem were added to the introduction, and the theoretical part was also modified. Additional analyzes were added to the results section.

Round 2

Reviewer 2 Report

Figure 3: x-axis is not clear. A grouped bar chart would be ideal for representing the results. 

In figure 5, the axes for the number of students and number of primary schools need to be included. Simplifying this figure by a stacked bar to represent the proportion and creating a new figure for schools would be ideal unless they are of the same scale. 

The research survey is too simple. Having a scale and adding more questions related to the factors resulting in satisfaction/dissatisfaction can result in meaningful insights. 

Author Response

comment 1: Figure 3: x-axis is not clear. A grouped bar chart would be ideal for representing the results. 

A: We totally agree with you. This figure is the automatic figure for this type of comparison in the statistical program which we used for data processing.

comment 2: In figure 5, the axes for the number of students and number of primary schools need to be included. Simplifying this figure by a stacked bar to represent the proportion and creating a new figure for schools would be ideal unless they are of the same scale. 

A: Figure 5 has been modified based on the request.

comment: The research survey is too simple. Having a scale and adding more questions related to the factors resulting in satisfaction/dissatisfaction can result in meaningful insights. 

A: Slovakia has a fragmented residential structure, which in practice means that there are currently 2,890 municipalities on the territory of Slovakia. Although the analyzed territory is located near the capital, from the point of view of transport accessibility, the analyzed municipalities are not as attractive as municipalities located near transport hubs. Nevertheless, a larger part of the inhabitants of these municipalities participated in the survey.

The analysis was part of a survey that included the entire spectrum of questions related to the life of inhabitants in the municipality from the point of view of the provision of services in the territory. The part related to the availability of educational facilities was part of a wider range of questions related to the evaluation of services that municipalities provide for their residents (social services, health services, cultural services, etc.). The base of this part of the survey among inhabitants was the expression of satisfaction or dissatisfaction with the given services in the context of their further solution in the future or a comparison of the development of satisfaction/dissatisfaction with the services provided. In the conditions of Slovakia, strategic documents are created at the local level in the context of the EU program period, and municipalities are obliged by law to create strategic documents, which also include a survey of inhabitants of municipalities with provided services. Involving residents in public opinion polls is complicated at the local level.

Characteristic features are the low participation of residents in public matters at the level of Slovakia. This is also confirmed by the results of the last municipal elections in November 2022, when the total participation of inhabitants reached only 46.19%. And another negative is the fact that many inhabitants do not answer all the questions in surveys, or they do not fill them in correctly, so it is subsequently impossible to evaluate them.

The base of the conducted research was only the assessment of satisfaction or dissatisfaction with the services provided, in further research, we will also deal with individual factors that may have an impact on the evaluation.

Reviewer 4 Report

Although this version is better than the first, I simply fail to see the contribution of this work to local economic development.

Author Response

comment: Although this version is better than the first, I simply fail to see the contribution of this work to local economic development.

A: Until 1989, education in Slovakia was centralized and managed by the Ministry of Education as the central body of state administration for the field of educational policy. However, the decentralization that took place on 01/07/2022 in the area of education management brought changes to this area as well. As part of this process, competence and responsibility in certain areas of education were transferred from the state administration to the local self-government. Under the conditions of Slovakia, the municipality primarily fulfills the role of the founder in the field of education, which it carries out within the exercise of original competencies (kindergartens, elementary art schools, language schools at elementary schools and school facilities), as well as within the transferred exercise of state administration (primary schools).

As the founder, the municipality appoints and dismisses the directors of individual schools and school facilities, which it is the founder of, performs state administration at the first level in matters of endangering education and training and neglecting to care for the compulsory school attendance of pupils. In the schools in which he is the founder, he checks the quality of meals served in the school canteens and checks compliance with generally binding legal regulations in the field of upbringing and education.

Economic aspects within the framework of regional education are closely connected with the effective management of schools and have an impact on the socio-economic development of the territory. Kindergartens, elementary art schools, language schools, and school facilities under the jurisdiction of municipalities since 01/01/2005 are financed from share taxes. Funds are provided to the founders of primary schools from the state budget. These financial resources are provided by the state in the form of a state subsidy for the founders, who subsequently redistribute them to primary schools. The founder is obliged to distribute all normative financial resources among the schools of which he is the founder. These financial resources cannot be used to finance kindergartens and school facilities, as these schools and school facilities are financed from municipal tax revenue. And in this context, the existence of preschool and school facilities at the local level becomes an important part of the socio-economic development of the territory. As long as there are preschool and school facilities in the municipality, the number of inhabitants of the municipality with a permanent residence also increases. This is an important aspect of municipal financing. Under the conditions of Slovakia, municipalities are financed from the state budget based on the number of inhabitants with permanent residence. The more inhabitants, the more funds municipalities receive from the state budget in the form of a share tax.

From the total amount of funds that the state collects in the form of personal income tax, it redistributes 70% to the level of municipalities based on the number of people with permanent residence in the municipality. These revenues make up 80% of the total revenues of the municipalities, and since they are not earmarked, the municipalities use them precisely for their socioeconomic development. Therefore we added lines 702-723.